# Effects of Variable Transport Properties on Heat and Mass Transfer in MHD Bioconvective Nanofluid Rheology with Gyrotactic Microorganisms: Numerical Approach



Muhammad Awais [1,*], Saeed Ehsan Awan [2], Muhammad Asif Zahoor Raja [3], Nabeela Parveen [1], Wasim Ullah Khan [4], Muhammad Yousaf Malik [5] and Yigang He [4,*]

1 Department of Mathematics, Attock Campus, COMSATS University Islamabad, Attock 43600, Pakistan; nabeela.mpa2016@gmail.com
2 Department of Electrical and Computer Engineering, Attock Campus, COMSATS University Islamabad, Attock 43600, Pakistan; saeed.ehsan@cuiatk.edu.pk
3 Future Technology Research Center, National Yunlin University of Science and Technology, 123 University Road, Section 3, Douliou, Yunlin 64002, Taiwan; rajamaz@yuntech.edu.tw
4 School of Electrical Engineering, Automation Wuhan University, Wuhan 430072, China; kwasim814@whu.edu.cn
5 Department of Mathematics, College of Sciences, King Khalid University, Abha 61421, Saudi Arabia; drmymalik@hotmail.com
* Correspondence: awais@ciit-attock.edu.pk (M.A.); yghe1221@whu.edu.cn (Y.H.); Tel.: +92-57-9316330 (M.A.); +86-027-6877-4003 (Y.H.)

**Abstract:** Rheology of MHD bioconvective nanofluid containing motile microorganisms is inspected numerically in order to analyze heat and mass transfer characteristics. Bioconvection is implemented by combined effects of magnetic field and buoyancy force. Gyrotactic microorganisms enhance the heat and transfer as well as perk up the nanomaterials' stability. Variable transport properties along with assisting and opposing flow situations are taken into account. The significant influences of thermophoresis and Brownian motion have also been taken by employing Buongiorno's model of nanofluid. Lie group analysis approach is utilized in order to compute the absolute invariants for the system of differential equations, which are solved numerically using Adams-Bashforth technique. Validity of results is confirmed by performing error analysis. Graphical and numerical illustrations are prepared in order to get the physical insight of the considered analysis. It is observed that for controlling parameters corresponding to variable transport properties $c_2$, $c_4$, $c_6$, and $c_8$, the velocity, temperature, concentration, and bioconvection density distributions accelerates, respectively. While heat and mass transfer rates increases for convection parameter and bioconvection Rayleigh number, respectively.

**Keywords:** bionanofluid; magnetic field; microorganisms; heat and mass transfer

## 1. Introduction

Concerning the importance of heat transfer management in energy systems, several mechanisms leading to heat transfer development in nanofluids have been analyzed. Nanofluids, initially introduced by Choi [1], are solid–liquid suspended materials containing nanofibers or solid nanoparticles, which possess novel chemical and physical characteristics. In rapidly intensifying field of nanotechnology, major consequences are remarkably strong temperature dependence of the thermal conductivity and a threefold higher critical heat flux as compared to conventional fluids. Nanofluids have been treated as a heat transfer fluid for industrial applications, cooling of electronic equipment, heat exchangers, as future coolants for computers, biological sensors, etc. In order to investigate convective heat transfer of nanofluids comprehensively, Buongiorno [2] proposed a mathematical model by ignoring dispersions limitations and homogeneous phenomenon.

He explained the thermal properties of nanofluids and suggested that Brownian diffusion and thermophoresis are significant mechanisms in convective heat and mass transfer enhancement without turbulence. Then, he persisted to rewrite the conservation equations based on them. A lot of work has been published in this regard. Recently, Nawaz et al. [3] numerically studied influence of homogeneous and heterogeneous chemical reactions on heat and mass transport in magnetized partially ionized nanofluid by employing finite element technique. They observed that homogeneous chemical reaction possesses more considerable influences on concentration than the impact of heterogeneous chemical reaction. Rehman et al. [4] numerically analyzed effects of temperature-dependent viscosity and thermal conductivity on molecular theory of liquid-originated nanofluid flow. They explored nanofluid attributes through Brownian motion and thermophoresis. They concluded that velocity pattern modifies for large size of thermophoresis diffusion and turn down by the growing magnitude of fluid parameters for Vogel's and Reynold's model. Awais et al. [5] numerically studied impact of Arrhenius activation energy on hyperbolic tangent nanofluid rheology with nonlinear Boussinesq approximation and mixed convection. They adopted Buongiorno's model for nanoparticles convection and analyzed entropy generation. It was observed that diffusion parameters reduce entropy production. The aspects of thermal and solutal energy transfer in MHD Burger's nanofluid dynamics were discussed by Iqbal et al. [6]. They adopted Fourier's and Fick's law to examine the heat and mass transfer phenomena. Convective heat and mass transfer analysis in nanofluids with different geometries and nanofluid models has been considered as [7–10].

Since, nanoparticles are not self-impelled and moves only due to thermophoresis and Brownian motion, high concentration of nanoparticles may cause problems of stability and rheology in heat and mass transfer enhancement. An amalgamation of nanofluids and biotechnological mechanisms developed by directional swimming of motile microorganisms may offer fruitful results in such cases. Motile microorganisms are self-impelled and want to accumulate in the vicinity of upper fluid layer, which form a dense upper surface. Their collective motion in nanofluid generates density gradient, which leads to convective instability and convection patterns in nanofluid and the phenomena is known as nanofluid bioconvection as pioneered by Kuznetsov [11]. These motile microorganisms can be classified according to the cause of implement to gyrotactic, oxytactic, gravitaxis, and chemotaxis. Bioconvection prevents agglomeration of nanoparticles and increase mass transfer in fluids. Wang and Fan [12] suggested that nanofluids contain micro and macro molecular mixture. Bioconvection corresponds to macroscopic convection of nanofluid produced by the density stratification. Bioconvection applications include biomicrosystems, the pharmaceutical industry, biological polymer synthesis, sustainable fuel cell technologies, microbial enhanced oil recovery, and continuous refinements in mathematical modelling. Since magnetic fields can highly affect the heat and mass transport processes in electrically conducting fluid, MHD bioconvection is an emerging field of interest and applicable in treatment of hyperthermia and arterial diseases, drawing of copper wires, and reduction of blood flow in surgeries and others. Three-dimensional flow of Maxwell nanofluid in coexistence of gyrotactic microorganisms with generative and absorptive heat transfer was examined by Ali et al. [13]. Sulaiman et al. [14] inspected 3D flow of Oldroyd-B nanofluid having gyrotactic microorganisms to analyze heat and mass exchange characteristics. The phenomenon of convection fortified by oxytactic microorganisms contains cell and transport the oxygen to the channel's lower part. Waqas et al. [15] investigated numerically the bioconvection dynamics of modified second-grade nanofluid in presence of nanoparticles and gyrotactic microorganisms. It was found that velocity of fluid reduces effectively due to bioconvection Rayleigh number and buoyancy ratio parameter. Awais et al. [16] exemplified the influence of variable heat immersion and stratification on heat and mass transfer in bioconvective rheology of nanomaterial containing gyrotactic microorganisms with inclined magnetic field effects. More work in this direction is cited as [17–20].

Flow dynamics due to moving surfaces have important role in the electrochemistry and manufacturing of polymers. The ejection of molten polymers from a slit die plays a

key role for the preparation of plastic sheets. Further applications include crystal growing, wire drawing, hot rolling, etc. Moreover, manufacturing of glass and production of paper are the important applications of such flow dynamics systems. Convective third-grade non-Newtonian fluid flow on a stretched sheet was analyzed by Rashidi et al. [21] for entropy optimization analysis. Ali et al. [22] investigated heat transfer properties in transport of mixed convection steady third-grade fluid over a resistant stretching cylinder with the influence of heat generation. They deduced that mixed convection parameter reduces the flow rate, while it increases for larger curvature of the cylinder. Awais et al. [23] numerically explored heat and mass transfer on the MHD Casson fluid flow from porous medium caused by shrinking surface associated to Lorentz force as well as heat generation and absorption effects and found dual numerical solutions for flow variables. It was noticed that both the temperature and concentration fields decline with Prandtl number. Mumraiz et al. [24] numerically analyzed influence of variable heat flux on MHD hybrid nanofluid flow caused by permeable stretching sheet with entropy optimization. It is concluded that suction and magnetic field reduce fluid flow, while it amplifies for rising magnitude of electric field, which leads to determine sticky effects. Ali et al. [25] carried out heat transfer analysis of Cu–Al$_2$O$_3$ hybrid nanofluid with heat flux and viscous dissipation. The review of literature shows that the analysis made for the three-dimensional flow past a stretching surface are more decline when heat and mass transport phenomena combined with other important features such as internal heat generation and absorption, thermal radiations, thermal diffusion effects, diffusion thermo effects, nanofluidics, and variable transport properties are considered. For instance, Hayat and Awais [26] investigated the 3D flow dynamics of an upper-convected Maxwell fluid and inspected the effects of Deborah number on the velocity components. Three-dimensional flow of Maxwell fluid because of an exponentially stretching sheet was scrutinized by Awais et al. [27]. They have adopted 3-stage Lobatto IIIA formula for numerical solutions and investigated that Deborah number retards the velocity. Shehzad et al. [28] studied MHD 3D flow of Maxwell liquid using Cattaneo–Christove heat and mass flux model with chemical reaction. Bilal et al. [29] recently presented the influences of temperature-dependent conductivity as well as absorptive and generative transport of heat on MHD 3D flow of Williamson fluid generated by bidirectional stretching surface. It was elucidated that velocity of the fluid shows opposite trend in the two directions of sheet, while nonlinearity index yields decrement in the velocity and thermal distributions of fluid. Three-dimensional flow caused by stretching surfaces with additional physical factors have been investigated by many authors and can be found in literatures [30–35].

Above-quoted literature reveals that research attempts pertaining to 3D flow dynamics of MHD bioconvection nanofluid containing gyrotactic microorganisms combined with variable transport properties and numerical solutions of mathematical model are scant. Our aim in this work is to nominate the 3D flow dynamics into new directions, and salient feature of the proposed scheme is briefly narrated as follows:

A novel investigation has been presented for bioconvection phenomenon involving gyrotactic microorganisms in three-dimensional flow dynamics of nanofluid.

Variable transport properties and assisting and opposing flow situations combined with magnetic field properties are incorporated in the model.

Mathematical modeling is performed by utilization of the conservation laws of mass, momentum, energy, mass fraction, and bioconvection processes along with suitable scaling procedure for the construction of system of differential equations.

Lie group analysis approach is presented to compute the absolute invariants for the differential system along with error analysis for validation of computed results.

Graphical and numerical illustrations are prepared in order to present the physical insight of the considered analysis.

Rest of the paper is arranged as follows: the problem formulation of the system is presented in Section 2, numerical procedure is introduced in Section 3 along with error

analysis, results with necessary discussion are provided in Section 4, while conclusions with future recommended studies are given in Section 5.

## 2. Model Formulation

Consider the three-dimensional rheology of bionanofluid with variable transport characteristics in the presence of microorganisms. Velocity field for 3D flow is $V = [\overline{u}(\overline{x}, \overline{y}), \overline{v}(\overline{x}, \overline{y}), \overline{w}(\overline{x}, \overline{y})]$. The velocities of a sheet in $\overline{x}$ and $\overline{y}$ dimensions are $\overline{u}_w(\overline{x}) = a\overline{x}$ and $\overline{v}_w(\overline{y}) = a\overline{y}$, where $a$ is any positive constant. Applied magnetic field of strength $B_0$ is considered by neglecting the induced magnetic field and electric field. Also, assume that suspended nanoparticles do not alter velocity and swimming direction of the motile microorganisms. The dimensional rheological model that governs the flow is given by:

$$\frac{\partial \overline{u}}{\partial \overline{x}} + \frac{\partial \overline{v}}{\partial \overline{y}} + \frac{\partial \overline{w}}{\partial \overline{z}} = 0, \tag{1}$$

$$\overline{u}\frac{\partial \overline{u}}{\partial \overline{x}} + \overline{v}\frac{\partial \overline{u}}{\partial \overline{y}} + \overline{w}\frac{\partial \overline{u}}{\partial \overline{z}} = \frac{1}{\rho_\infty}\frac{\partial}{\partial \overline{z}}\left[\mu(C)\frac{\partial \overline{u}}{\partial \overline{z}}\right] - \frac{\sigma B_0^2 \overline{u}}{\rho_\infty} + (1 - C_\infty)Bg(T - T_\infty) \\ - \frac{1}{\rho_\infty}\left[\left(\rho_g - \rho_\infty\right)g(C - C_\infty) + (\rho_m - \rho_\infty)g\gamma(n - n_\infty)\right], \tag{2}$$

$$\overline{u}\frac{\partial \overline{v}}{\partial \overline{x}} + \overline{v}\frac{\partial \overline{v}}{\partial \overline{y}} + \overline{w}\frac{\partial \overline{v}}{\partial \overline{z}} = \frac{1}{]\rho_\infty}\frac{\partial}{\partial \overline{z}}\left[\mu(C)\frac{\partial \overline{v}}{\partial \overline{z}}\right] - \frac{\sigma B_0^2 \overline{v}}{\rho_\infty}, \tag{3}$$

$$\overline{u}\frac{\partial T}{\partial \overline{x}} + \overline{v}\frac{\partial T}{\partial \overline{y}} + \overline{w}\frac{\partial T}{\partial \overline{z}} = \tau'\frac{\partial}{\partial \overline{z}}[D_B(C)C]\frac{\partial T}{\partial \overline{z}} + \frac{1}{\rho_\infty c_p}\frac{\partial}{\partial \overline{z}}\left[k(C)\frac{\partial T}{\partial \overline{z}}\right] + \tau'\frac{D_T}{T_\infty}\left(\frac{\partial T}{\partial \overline{z}}\right)^2, \tag{4}$$

$$\overline{u}\frac{\partial C}{\partial \overline{x}} + \overline{v}\frac{\partial C}{\partial \overline{y}} + \overline{w}\frac{\partial C}{\partial \overline{z}} = \frac{D_T}{T_\infty}\frac{\partial^2 T}{\partial \overline{z}^2} + \frac{\partial}{\partial \overline{z}}\left[D_B(C)\frac{\partial C}{\partial \overline{z}}\right], \tag{5}$$

$$\overline{u}\frac{\partial n}{\partial \overline{x}} + \overline{v}\frac{\partial n}{\partial \overline{y}} + \overline{w}\frac{\partial n}{\partial \overline{z}} = \frac{\partial}{\partial \overline{z}}\left[D_n(C)\frac{\partial n}{\partial \overline{z}}\right] - \frac{bW_C}{C_w - C_\infty}\left[\frac{\partial}{\partial \overline{z}}\left(n\frac{\partial C}{\partial \overline{z}}\right)\right], \tag{6}$$

where $(\overline{x}, \overline{y}, \overline{z})$ and $(\overline{u}, \overline{v}, \overline{w})$ represent dimensions and velocity component in Cartesian coordinate system, respectively. Moreover, $T$ is fluid temperature, $C$ is nanoparticles' concentration, and $n$ is fraction of motile microorganisms.

Corresponding no-slip wall properties are:

$$\overline{u} = \overline{u}_w(\overline{x}) = a\overline{x},\ \overline{v} = \overline{v}_w(\overline{y}) = a\overline{y},\ \overline{w} = 0,\ T = T_w,\ C = C_w,\ n = n_w\ at\ \overline{z} = 0, \\ \overline{u} = 0, \overline{v} = 0, T \to T_\infty, C \to C_\infty, n \to n_\infty\ \text{as}\ \overline{z} \to \infty. \tag{7}$$

The variable thermophysical quantities are given as:

$$\mu(C) = [\mu_\infty + c_1\mu_\infty(C - C_\infty)] = \mu_\infty + c_2\mu_\infty\phi, \\ \kappa(C) = [\kappa_\infty + c_3\kappa_\infty(C - C_\infty)] = \kappa_\infty + c_4\kappa_\infty\phi, \\ D_B(C) = [D_{B,\infty} + c_5 D_{B,\infty}(C - C_\infty)] = D_{B,\infty} + c_6 D_{B,\infty}\phi, \\ D_n(C) = [D_{n,\infty} + c_7 D_{n,\infty}(C - C_\infty)] = D_{n,\infty} + c_8 D_{n,\infty}\phi. \tag{8}$$

We establish nondimensional variables and transformations as follows:

$$x * L = \overline{x}, y * L = \overline{y}, z * L = \overline{z}, u = \frac{1}{\sqrt{av_\infty}}(\overline{u}), v = \frac{1}{\sqrt{av_\infty}}(\overline{v}), w = \frac{1}{\sqrt{av_\infty}}(\overline{w}), u_e = \frac{\overline{u}_e}{\sqrt{av_\infty}}, \\ v_e = \frac{\overline{v}_e}{\sqrt{av_\infty}}, u_w = \frac{\overline{u}_w}{\sqrt{av_\infty}}, \theta = \frac{T - T_\infty}{T_f - T_\infty}, \phi = \frac{C - C_\infty}{C_W - C_\infty}, \xi = \frac{n}{n_w}, u^* e^{-\varepsilon\alpha_1} = u, v^* e^{-\varepsilon\alpha_2} = v, \\ w^* e^{-\varepsilon\alpha_3} = w, x^* e^{-\varepsilon\alpha_4} = x, y^* e^{-\varepsilon\alpha_5} z^* e^{-\varepsilon\alpha_6} = z, u_e^* e^{-\varepsilon\alpha_7} = u_e, v_e^* e^{-\varepsilon\alpha_8} = v_e, \phi^* e^{-\varepsilon\alpha_9} = \phi, \\ \theta^* e^{-\varepsilon\alpha_{10}} = \theta, \xi^* e^{-\varepsilon\alpha_{11}} = \xi, u_w^* e^{-\varepsilon\alpha_{12}} = u_w, v_w^* e^{-\varepsilon\alpha_{13}} = v_w, \eta = z, u = x * f'(\eta), v = y * g'(\eta), \\ \theta = \theta(\eta), \phi = \phi(\eta), w = -f(\eta) - g(\eta), \xi = \xi(\eta), u_e = x, u_w = x(u_w)_0, v_e = y, v_w = x(v_w)_0.$$

We get the following set of ODEs:

$$(1 + c_2\phi)\frac{d^3 f}{d\eta^3} + (f + g)\frac{d^2 f}{d\eta^2} - \left(\frac{df}{d\eta}\right)^2 + c_2\frac{d^2 f}{d\eta^2}\frac{d\phi}{d\eta} - M^2\frac{df}{d\eta} + \lambda(\theta - Nr\phi - Rb\xi) = 0, \tag{9}$$

$$(1 + c_2\phi)\frac{d^3g}{d\eta^3} + (f + g)\frac{d^2g}{d\eta^2} - \left(\frac{dg}{d\eta}\right)^2 + c_2\frac{d^2g}{d\eta^2}\frac{d\phi}{d\eta} - M^2\frac{dg}{d\eta} = 0, \tag{10}$$

$$(1 + c_4\phi)\frac{d^2\theta}{d\eta^2} + \Pr(f + g)\frac{d\theta}{d\eta} + Nt\left(\frac{d\theta}{d\eta}\right)^2 + c_4\frac{d\theta}{d\eta}\frac{d\phi}{d\eta} + Nb\frac{d\theta}{d\eta}\frac{d\phi}{d\eta}[1 + 2c_6\phi + c_6] = 0, \tag{11}$$

$$(1 + c_6\phi)\frac{d^2\phi}{d\eta^2} + c_6\left(\frac{d\phi}{d\eta}\right)^2 + Sc(f + g)\frac{d\phi}{d\eta} + \frac{Nt}{Nb}\frac{d^2\theta}{d\eta^2} = 0, \tag{12}$$

$$(1 + c_6\phi)\frac{d^2\phi}{d\eta^2} + c_6\left(\frac{d\phi}{d\eta}\right)^2 + Sc(f + g)\frac{d\phi}{d\eta} + \frac{Nt}{Nb}\frac{d^2\theta}{d\eta^2} = 0, \tag{13}$$

Associated dimensionless boundary conditions get the form:

$$(1 + c_6\phi)\frac{d^2\phi}{d\eta^2} + c_6\left(\frac{d\phi}{d\eta}\right)^2 + Sc(f + g)\frac{d\phi}{d\eta} + \frac{Nt}{Nb}\frac{d^2\theta}{d\eta^2} = 0, \tag{14}$$

Dimensionless physical parameters are mathematically expressed as follows:

$$\begin{aligned}
M &= \sqrt{\frac{\sigma B_0^2}{a\rho_\infty}}, \lambda = \frac{gB\rho_\infty(1 - C_\infty)(T_w - T_\infty)}{a\rho_\infty u_w}, Nr = \frac{(\rho_p - \rho_\infty)(C_w - C_\infty)}{B\rho_\infty(1 - C_\infty)(T_w - T_\infty)}, \\
\Pr &= \frac{\mu_\infty c_p}{\kappa_\infty}, Rb = \frac{\gamma(\rho_m - \rho_\infty)(n_w - n_\infty)}{B\rho_\infty(1 - C_\infty)(T_w - T_\infty)} Nb = \frac{\rho_\infty c_p \tau' D_{B,\infty}(C_w - C_\infty)}{\kappa_\infty}, \\
Nt &= \frac{\rho_\infty c_p \tau' D_T(T_w - T_\infty)}{\kappa_\infty T_\infty}, \mathrm{Re} = \frac{u_w L}{\nu_\infty}, Lb = \frac{\nu_\infty}{D_n}, Sc = \frac{\nu_\infty}{D_{B,\infty}}, Pe = \frac{bW_c}{D_n}.
\end{aligned} \tag{15}$$

The physical quantities in this study are the skin friction along the $x$-axis $C_{f_{\overline{x}}}$, skin friction along $y$-axis $C_{f_{\overline{y}}}$, Nusselt number $Nu_{\overline{x}}$, Sherwood number $Sh_{\overline{x}}$, and the density number of motile microorganisms $Nn_{\overline{x}}$ defined as:

$$\begin{aligned}
C_{f_{\overline{x}}} &= \frac{\tau_{w\overline{x}}}{\rho\overline{u}_e^2}, C_{f_{\overline{y}}} = \frac{\tau_{w\overline{y}}}{\rho\overline{v}_e^2}, Nu_{\overline{x}} = \frac{\overline{x}q_w}{\kappa(C)(T_f - T_\infty)}, \\
Sh_{\overline{x}} &= \frac{\overline{x}q_m}{D_B(C)(C_w - C_\infty)}, Nn_{\overline{x}} = \frac{\overline{x}q_n}{D_n(C)(n_w - n_\infty)}.
\end{aligned}$$

where important physical quantities of the fluidic problems are given as:

$$\tau_{w\overline{x}} = \mu(C)\frac{\partial\overline{u}}{\partial\overline{z}}\bigg|_{\overline{z}=0}, \tau_{w\overline{y}} = \mu(C)\frac{\partial\overline{v}}{\partial\overline{z}}\bigg|_{\overline{z}=0}, q_m = -D_B(C)\frac{\partial C}{\partial\overline{z}}\bigg|_{\overline{z}=0}, q_n = -D_n(C)\frac{\partial\xi}{\partial\overline{z}}\bigg|_{\overline{z}=0}. \tag{16}$$

The dimensionless forms of abovementioned quantities are:

$$\begin{aligned}
\mathrm{Re}\overline{x}^{\frac{1}{2}}C_{f\overline{x}} &= (1 + c_2\phi(0))f''(0), \mathrm{Re}\overline{y}^{\frac{1}{2}}C_{f\overline{y}} = (1 + c_2\phi(0))g''(0), \\
\mathrm{Re}\overline{x}^{\frac{-1}{2}}Nu_{\overline{x}} &= -\theta'(0), \mathrm{Re}\overline{x}^{\frac{-1}{2}}Sh_{\overline{x}} = -\frac{\phi'(0)}{\phi(0)}, \mathrm{Re}\overline{x}^{\frac{-1}{2}}Nn_{\overline{x}} = -\xi'(0).
\end{aligned} \tag{17}$$

## 3. Numerical Procedure

The competency of Adam Numerical Solver based on predictor-corrector approach [36–42] is exploited for numerical solution. Equations (9)–(13) are stiff nonlinear, therefore, numerical treatment is conducted with Adams method. The workflow diagram of the proposed methodology is presented in Figure 1.

**1. Problem Formulation**

Fluid dynamics models
Bio-nanofluidics systems

3D-Convective Rheology of Bio-nanofluidic system in the presence of Micro-Organism

**2. Mathematical Modeling**

**System of ODEs**

$$(1+c_2\phi)\frac{d^3f}{d\eta^3}+(f+g)\frac{d^2f}{d\eta^2}-\left(\frac{df}{d\eta}\right)^2+c_2\frac{d^2f}{d\eta^2}\frac{d\varphi}{d\eta}-$$

$$M^2\frac{df}{d\eta}+\lambda(\theta-Nr\phi-Rb\xi)=0$$

$$(1+c_6\phi)\frac{d^2\varphi}{d\eta^2}+c_6\left(\frac{d\varphi}{d\eta}\right)^2+Sc(f+g)\frac{d\varphi}{d\eta}+\frac{Nt}{Nb}\frac{d^2\theta}{d\eta^2}=0$$

**3. Dynamics of System**

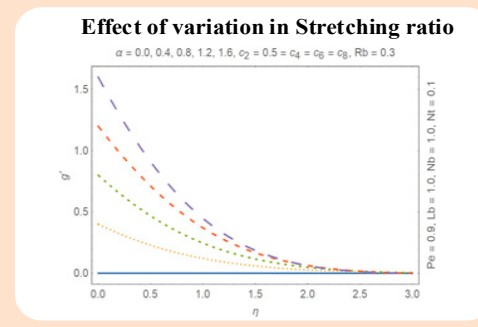

**4. Analysis**

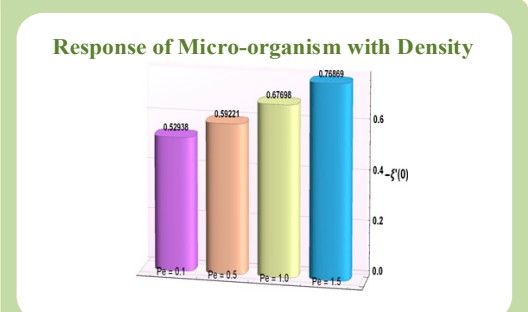

**5. Comparative Analysis**

**Analysis of accuracy**

Absolute error bases analysis for numerical and analytical solvers

Analysis of the flow by variation of Sherwood and Nusselt numbers

**Figure 1.** Block structure representation of workflow of the system.

### 3.1. Adams Method: Predictor-Corrector Approach

The magneto-hydrodynamic three-dimensional bioconvection rheology of nanomaterial involving gyrotactic microparticles is represented in Equations (9)–(13) and are transformed in to equivalent first-order system along with boundary conditions in terms of velocity field along $x$-axis $f(\eta)$, velocity field along $y$-axis $g(\eta)$, temperature profile $\theta(\eta)$, concentration profile $\phi(\eta)$, and density profile $\xi(\eta)$.

First-order system for $x$-component of velocity field $f(\eta)$ is formulated as:

$$f_1 = f', f_2 = f'_1,$$
$$f_3 = f'_2 = \left(\frac{1}{1+c_2\phi}\right)\left[-f_2(f+g)+(f_1)^2-c_2f_2\phi_1+M^2f_1-\lambda(\theta-Nr\phi-Rb\xi)\right]. \tag{18}$$

Similarly, first-order system for y-component velocity field $g(\eta)$ is formulated as:

$$g_1 = g', g_2 = g'_1,$$
$$g_3 = g_2' = \left(\frac{1}{1+c_2\phi}\right)\left[-g_2(f+g)+g_1^2-c_2g_2\phi_1+M^2g_1\right]. \tag{19}$$

Moreover, first-order system for temperature profile $\theta(\eta)$ will take following form:

$$\theta_1 = \theta',$$
$$\theta_2 = \theta_1' = \left(\frac{1}{1+c_4\phi}\right)\left[-\Pr(f+g)\theta_1 - Nt\theta_1^2 - c_4\theta_1\phi_1 - Nb\theta_1\phi_1(1+c_6(1+2\phi))\right].$$

(20)

In addition, first-order system for the case of concentration profile $\phi(\eta)$ will be:

$$\phi_1 = \phi',$$
$$\phi_2 = \phi'_1 = \left(\frac{1}{1+c_6\phi}\right)\left[-c_6\phi_1^2 - Sc(f+g)]\phi_1 - \frac{Nt}{Nb}\theta_2\right].$$

(21)

Furthermore, finally first-order system for density profile $\xi(\eta)$ is as follows:

$$\xi_1 = \xi',$$
$$\xi_2 = \xi_1' = \left(\frac{1}{1+c_8\phi}\right)[-Sb(f+g)\xi_1 + Pe\xi\phi_2 - (c_8 - Pe)\phi_1\xi_1].$$

(22)

The Boundary conditions for $f(\eta)$, $g(\eta)$, $\theta(\eta)$, $\phi(\eta)$ and $\xi(\eta)$ are given, respectively, as follows:

$$f(0) = 0, f_1(0) = 1, f_1(\infty) = 0, g(0) = 0, g_1(0) = \alpha, g_1(\infty) = 0,$$
$$\theta(0) = 1, \theta(\infty) = 0, \phi(0) = 1, \phi(\infty) = 0, \xi(0) = 1, \xi(\infty) = 0.$$

(23)

Generic representation of derived differential system for $f(\eta)$, $g(\eta)$, $\theta(\eta)$, $\phi(\eta)$, and $\xi(\eta)$ is given, respectively, as follows:

$$\frac{df}{d\eta} = s(\eta, f), f(\eta_0) = f_0,$$

(24)

$$\frac{dg}{d\eta} = s(\eta, g), g(\eta_0) = g_0,$$

(25)

$$\frac{d\theta}{d\eta} = s(\eta, \theta), \theta(\eta_0) = \theta_0,$$

(26)

$$\frac{d\phi}{d\eta} = s(\eta, \phi), \phi(\eta_0) = \phi_0,$$

(27)

$$\frac{d\xi}{d\eta} = s(\eta, \xi), \xi(\eta_0) = \xi_0,$$

(28)

Solution can be obtained by transforming the equation into first-order expression utilizing two stages Adams-Bashforth solver for $f(\eta)$, $g(\eta)$, $\theta(\eta)$, $\phi(\eta)$, and $\xi(\eta)$, which is given as follows:

$$f_{k+1} = f_k + \frac{h}{2}(3s(\eta_k, f_k) - s(\eta_{k-1}, f_{k-1})),$$

(29)

$$g_{k+1} = g_k + \frac{h}{2}(3s(\eta_k, g_k) - s(\eta_{k-1}, g_{k-1})),$$

(30)

$$\theta_{k+1} = \theta_k + \frac{h}{2}(3s(\eta_k, \theta_k) - s(\eta_{k-1}, \theta_{k-1})),$$

(31)

$$\phi_{k+1} = \phi_k + \frac{h}{2}(3s(\eta_k, \phi_k) - s(\eta_{k-1}, \phi_{k-1})),$$

(32)

$$\xi_{k+1} = \xi_k + \frac{h}{2}(3s(\eta_k, \xi_k) - s(\eta_{k-1}, \xi_{k-1})),$$

(33)

where $h$ represents step size parameter.

Similarly, generalized two-stage Adams-Moulton corrector expressions [36–42] for $f(\eta), g(\eta), \theta(\eta), \phi(\eta)$, and $\xi(\eta)$ are given as follows:

$$f_{k+1} = f_k + \frac{h}{2}(s(\eta_{k+1}, f_{k+1}) - s(\eta_k, f_k)), \tag{34}$$

$$g_{k+1} = g_k + \frac{h}{2}(s(\eta_{k+1}, g_{k+1}) - s(\eta_k, g_k)), \tag{35}$$

$$\theta_{k+1} = \theta_k + \frac{h}{2}(s(\eta_{k+1}, \theta_{k+1}) - s(\eta_k, \theta_k)), \tag{36}$$

$$\phi_{k+1} = \phi_k + \frac{h}{2}(s(\eta_{k+1}, \phi_{k+1}) - s(\eta_k, \phi_k)), \tag{37}$$

$$\xi_{k+1} = \xi_k + \frac{h}{2}(s(\eta_{k+1}, \xi_{k+1}) - s(\eta_k, \xi_k)), \tag{38}$$

Accordingly, the expression for higher order Predictor-Corrector based Adam Method can be constructed, however, in our study, we used Mathematica Built in routine "NDSolve" with algorithm "Adams" for numerical treatment of the system.

### 3.2. Finite Difference Method

The discretization formulas for central finite difference method based on 7-point stencils for density profile $\xi(\eta)$ are specified as follows:

$$\xi'(\eta) = \frac{-\xi(\eta - 3h) + 9\xi(\eta - 2h) - 45\xi(\eta - h) + 45\xi(\eta + h) - 9\xi(\eta + 2h) + \xi(\eta + 3h)}{60h}, \tag{39}$$

$$\xi''(\eta) = \frac{\xi(\eta - 3h) - \frac{27}{2}\xi(\eta - 2h) + 135\xi(\eta - h) - 245\xi(\eta) + 135\xi(\eta + h) - \frac{27}{2}\xi(\eta + 2h) + \xi(\eta + 3h)}{90h^2}, \tag{40}$$

$$\xi'''(\eta) = \frac{\xi(\eta - 3h) - 8\xi(\eta - 2h) + 13\xi(\eta - h) - 13\xi(\eta + h) + 8\xi(\eta + 2h) - \xi(\eta + 3h)}{8h^3}, \tag{41}$$

In the same manner, the forward and backward difference formulas [43,44] for density profile $\xi(\eta)$ can be defined, which are further computed by the procedure of standard iterative. In the same way, we can construct discretization formulas for $f(\eta), g(\eta), \theta(\eta)$, and $\phi(\eta)$, respectively.

### 3.3. Error Analysis

Numerical solutions have been developed for transformed governing equations by adopting Adams-Bashforth predictor-corrector method and finite difference method. In order to test convergence and stability of these methods, error analysis have been prepared and portrayed in Figure 2a,f for dimensionless magnetic and thermophoretic parameters. It is detected from error plots that absolute errors in flow profiles for salient parameters are quite negligible. Figure 2a,d depicts the absolute error of magnetic parameter with velocity profile, concentration profile, and density profile, and it lies in the range of $10^{-09}$ to $10^{-06}$, which is quite negligible. While absolute error of thermophoretic parameter with velocity profile, concentration profile, and density profile are represented in Figure 2e,f. These plots show that obtained solutions are convergent and satisfy the tolerance level of $10^{-8}$.

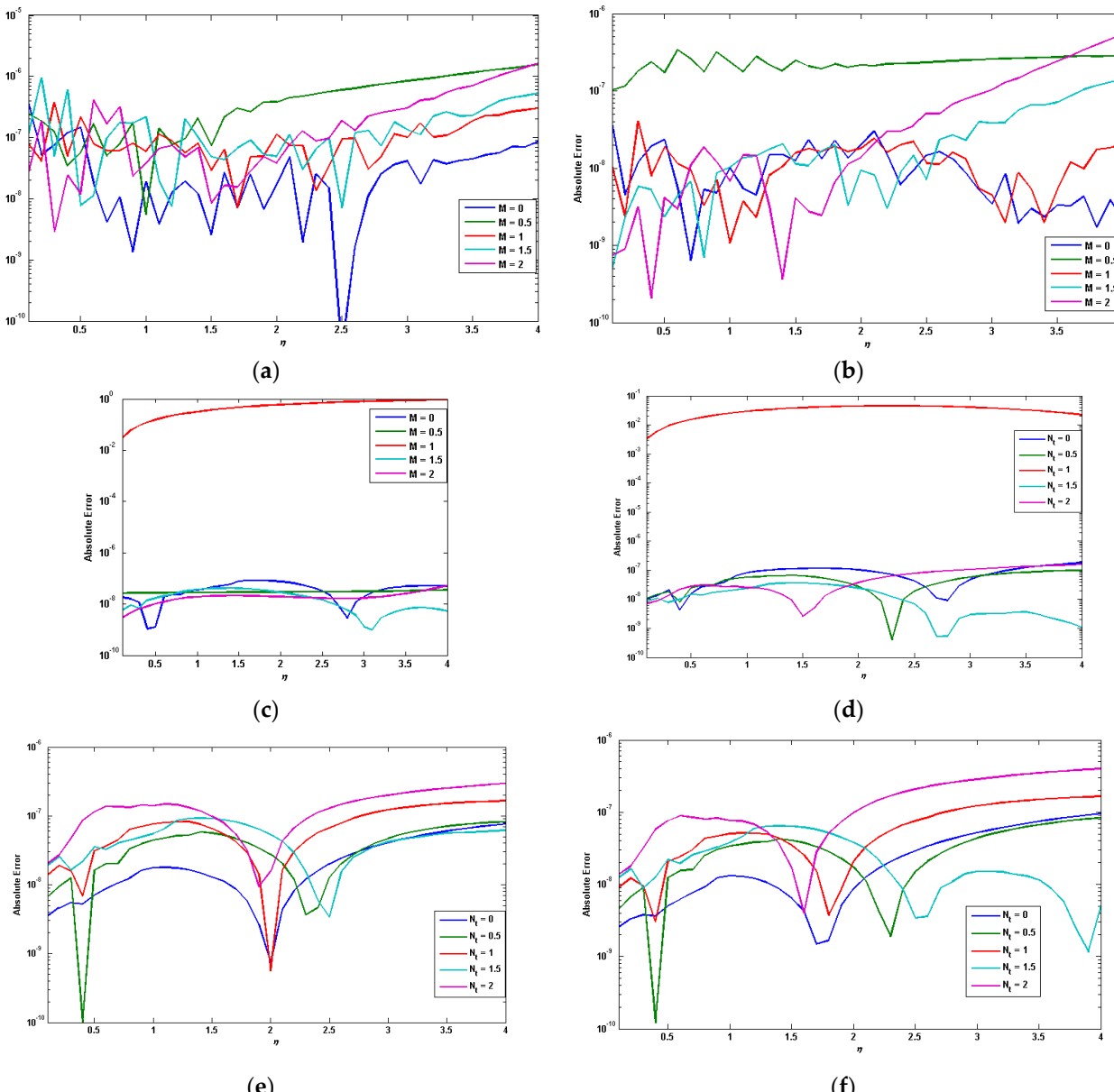

**Figure 2.** Behavior of absolute error in (**a**) $f'(\eta)$ for magnetic parameter, (**b**) $\phi(\eta)$ for magnetic parameter, (**c**) $\xi(\eta)$ for magnetic parameter, (**d**) $\theta(\eta)$ for thermophoretic parameter, (**e**) $\phi(\eta)$ for thermophoretic parameter, and (**f**) $\xi(\eta)$ for thermophoretic parameter.

## 4. Results and Discussion

In this portion, we have disclosed the physical interpretation of sundry parameters emerging in the model. The response of velocity profile is displayed in Figure 3a,d. The impact of viscosity variation parameter $c_2$ on velocity field $f'(\eta)$ is plotted in Figure 3a. Positive values of $c_2$ yield higher temperature difference between sheet and ambient fluid. Physical significance is reduction in viscosity of the fluid with an enhancement in temperature due to weak intermolecular forces, which leads to a gradual increment in fluid velocity. It is noticed from Figure 3b that $f'(\eta)$ declines for higher values of bioconvection Rayleigh number *Rb*. It is due to the fact that large values of *Rb* intensify the buoyancy forces and increase the concentration of both nanoparticles and microorganisms thus weaken the convection of fluid. The variation in $f'(\eta)$ for escalating values of magnetic parameter *M* is incorporated in Figure 3c, which gives an asymptotic reduction in it. Physics behind this behavior is effect of strong Lorentz force and very slow induction in electrically conducting fluid generated by applied magnetic field with small magnetic

Reynolds number, which result in an increment in frictional effects, dragging of the velocity, and, hence, decelerating of the flow. Figure 3d depicts the variation in y-component of velocity $g'(\eta)$ field against stretching ratio parameter $\alpha$. It is seen that for $\alpha = 0$, graph of $g'(\eta)$ approaches to zero, which indicates unidirectional stretching case. Moreover, for large values of $\alpha$, sheet velocity increases and thus velocity of nanofluid increases rapidly due to no-slip wall conditions.

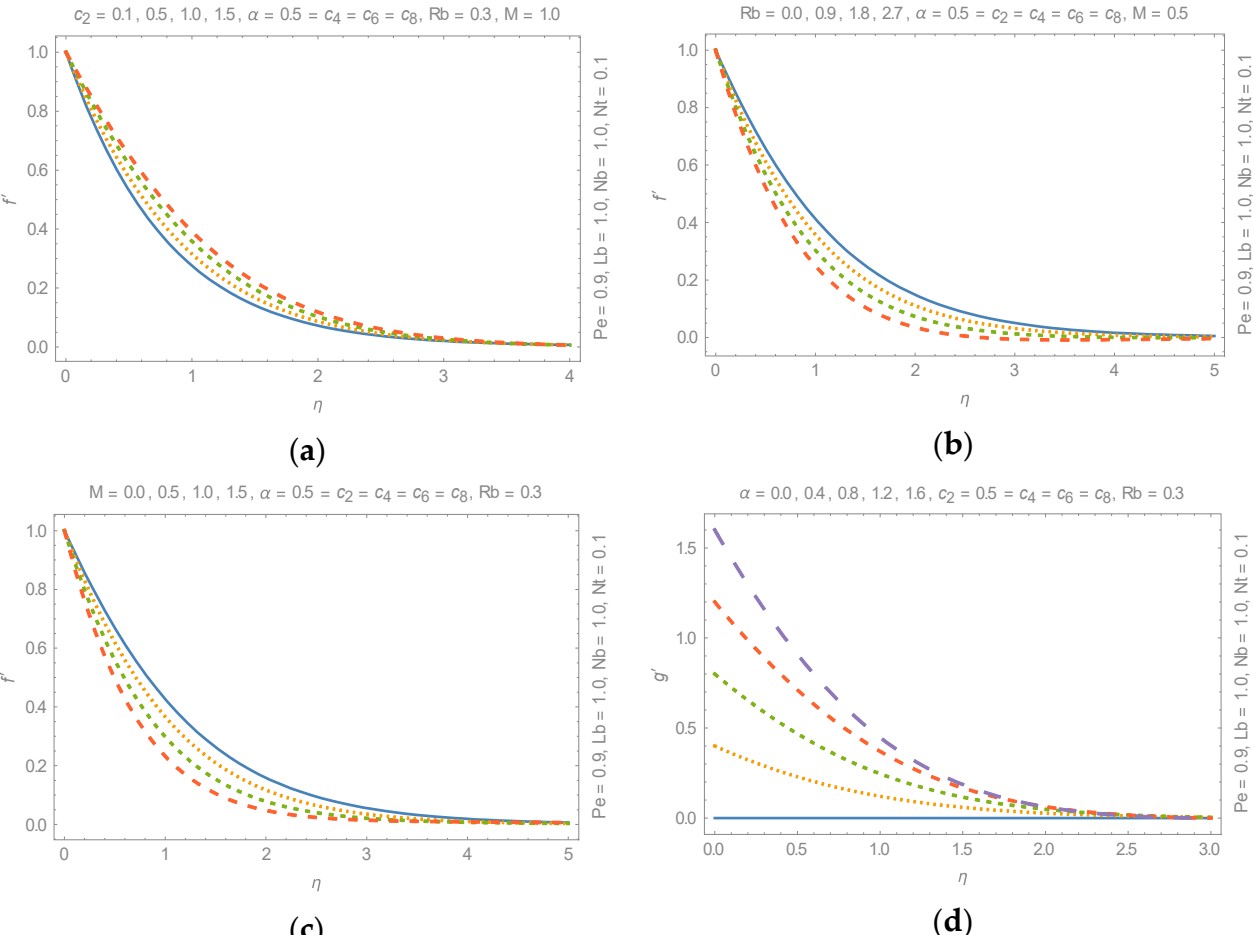

**Figure 3.** Behavior of (**a**) $f'(\eta)$ for $c_2$, (**b**) $f'(\eta)$ for $Rb$, (**c**) $f'(\eta)$ for $M$, and (**d**) $g'(\eta)$ for $\alpha$.

Moreover, positive values of thermal conductivity parameter $c_4$ specify high thermal conductivity of nanofluid, which increases thermal boundary layer thickness and temperature of fluid as explored in Figure 4a. It is seen in Figure 4b that temperature is an increasing function of *Nb*. This situation occurs physically because of random motion of nanoparticles for larger Brownian motion parameter and such collision produces additional heat. Thus, the expansion in temperature curves is observed. The response of dimensionless concentration field for several values of $c_6$ is reported in Figure 4c. We infer that upsurge in magnitude of $c_6$ augments the concentration boundary layer thickness due to increase in mass diffusivity and hence concentration profile rises. A similar trend is noted for microorganism diffusivity parameter $c_8$ on density profile $\xi(\eta)$. This observation occurs due to increment in thickness of microscopic swimmers as explicated in Figure 4d.

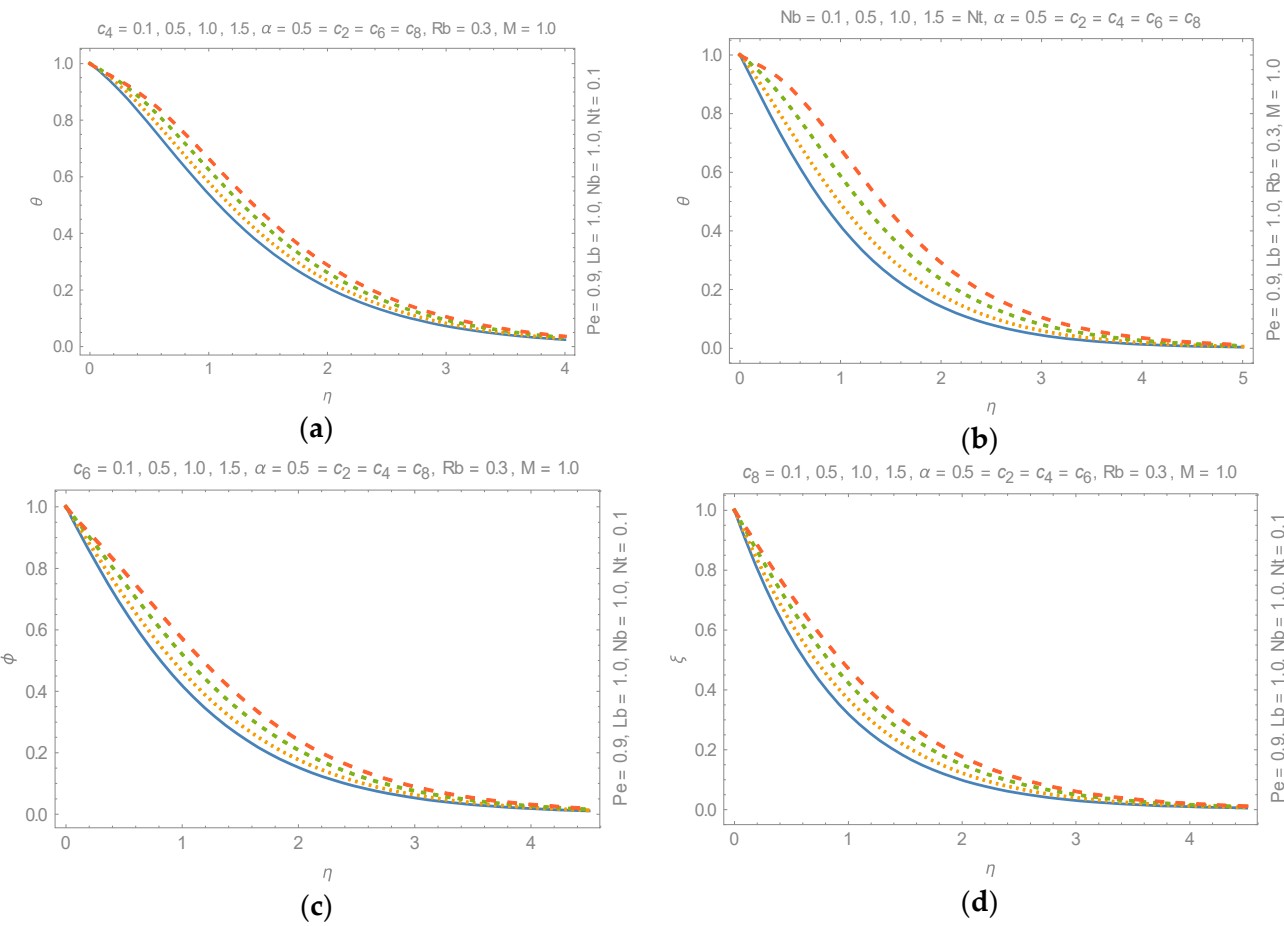

**Figure 4.** Behavior of (**a**) θ(η) for $c_4$, (**b**) θ(η) for $Nb$, (**c**) φ(η) for $c_6$, and (**d**) ξ(η) for $c_8$.

In addition, substantial quantities, e.g., $-f''(0)$ and $-g''(0)$ along $x$- and $y$- axes, heat flux $-\theta'(0)$, mass flux $-\phi'(0)$, and local density number $-\xi'(0)$ of motile microorganisms against variation in controlling parameters are estimated and exhibited in Tables 1 and 2. For a detailed view, bar charts have been drawn for augmented values of some of the incriminating parameters. The effects of assisting and opposing convection heat transfer on skin friction coefficient are elucidated in Figure 5a,b. It is seen comparatively that when no-slip wall conditions are imposed, assisting flow for $\lambda \succ 0$ declines surface friction, while in case of opposing flow for $\lambda \prec 0$, surface friction grows sensitively due to enhancing disturbance and irreversibility. Figure 6a depicts the variation in heat flux for both assisting and opposing flow and perceived that as the flow behavior varies from opposing to assisting, the rate of heat transfer rises occurred due to enhancing temperature difference. The variation in concentration field against $Rb$ is explicated in Figure 6b, and a gradually decreasing behavior of $-\phi'(0)$ is noticed. Further, the response of microorganisms' local density number for variation in $Pe$ and $Lb$ is represented in Figure 7a,b, respectively. As rise in $Pe$ indicates enhancement in advection transport rate compared to diffusion, accumulation of microorganisms close to the surface boosts due to which density number enlarges while a reverse trend is discerned for bioconvection Lewis number.

**Table 1.** Skin friction coefficient values and Nusselt number against significant parameters.

| M | $N_t$ | $N_b$ | $N_r$ | $R_b$ | $S_c$ | $L_b$ | $\lambda$ | $-f''(0)$ | $-g''(0)$ | $-\theta'(0)$ |
|---|---|---|---|---|---|---|---|---|---|---|
| 0.0 | 0.5 | 0.5 | 0.3 | 0.1 | 0.5 | 0.1 | 0.5 | 0.762800 | 0.05401 | 0.332037 |
| 0.5 | 0.5 | 0.5 | 0.3 | 0.1 | 0.5 | 0.1 | 0.5 | 1.168734 | 0.484402 | 0.000032 |
| 1.0 | 0.5 | 0.5 | 0.3 | 0.1 | 0.5 | 0.1 | 0.5 | 1.078827 | 0.096633 | 0.292369 |
| 1.5 | 0.5 | 0.5 | 0.3 | 0.1 | 0.5 | 0.1 | 0.5 | 1.393633 | 0.132271 | 0.259094 |
| 2.0 | 0.5 | 0.5 | 0.3 | 0.1 | 0.5 | 0.1 | 0.5 | 1.750901 | 0.170221 | 0.228761 |
| 1.0 | 0.0 | 0.5 | 0.3 | 0.1 | 0.5 | 0.1 | 0.5 | 1.066443 | 0.096007 | 0.321714 |
| 1.0 | 0.3 | 0.5 | 0.3 | 0.1 | 0.5 | 0.1 | 0.5 | 1.075074 | 0.096421 | 0.304447 |
| 1.0 | 0.6 | 0.5 | 0.3 | 0.1 | 0.5 | 0.1 | 0.5 | 1.080143 | 0.096722 | 0.286202 |
| 1.0 | 1.0 | 0.5 | 0.3 | 0.1 | 0.5 | 0.1 | 0.5 | 1.081972 | 0.096966 | 0.260988 |
| 1.0 | 0.5 | 0.3 | 0.3 | 0.1 | 0.5 | 0.1 | 0.5 | 1.102757 | 0.097133 | 0.336568 |
| 1.0 | 0.5 | 0.6 | 0.3 | 0.1 | 0.5 | 0.1 | 0.5 | 1.211876 | 0.114099 | 0.408964 |
| 1.0 | 0.5 | 1.0 | 0.3 | 0.1 | 0.5 | 0.1 | 0.5 | 1.211120 | 0.114538 | 0.2706306 |
| 1.0 | 0.5 | 0.5 | 0.0 | 0.1 | 0.5 | 0.1 | 0.5 | 1.163783 | 0.113966 | 0.451283 |
| 1.0 | 0.5 | 0.5 | 0.5 | 0.1 | 0.5 | 0.1 | 0.5 | 1.211073 | 0.113898 | 0.450755 |
| 1.0 | 0.5 | 0.5 | 1.0 | 0.1 | 0.5 | 0.1 | 0.5 | 1.258386 | 0.113830 | 0.450227 |
| 1.0 | 0.5 | 0.5 | 0.3 | 0.0 | 0.5 | 0.1 | 0.5 | 1.181687 | 0.113941 | 0.451091 |
| 1.0 | 0.5 | 0.5 | 0.3 | 0.5 | 0.5 | 0.1 | 0.5 | 1.234045 | 0.113861 | 0.450467 |
| 1.0 | 0.5 | 0.5 | 0.3 | 1.0 | 0.5 | 0.1 | 0.5 | 1.286456 | 0.113781 | 0.449842 |
| 1.0 | 0.5 | 0.5 | 0.3 | 0.1 | 0.0 | 0.1 | 0.5 | 1.196631 | 0.114316 | 0.457991 |
| 1.0 | 0.5 | 0.5 | 0.3 | 0.1 | 0.5 | 0.1 | 0.5 | 1.192154 | 0.113925 | 0.450966 |
| 1.0 | 0.5 | 0.5 | 0.3 | 0.1 | 1.0 | 0.1 | 0.5 | 1.187753 | 0.113541 | 0.444190 |
| 1.0 | 0.5 | 0.5 | 0.3 | 0.1 | 0.5 | 0.0 | 0.5 | 1.192189 | 0.113925 | 0.450965 |
| 1.0 | 0.5 | 0.5 | 0.3 | 0.1 | 0.5 | 0.5 | 0.5 | 1.192015 | 0.113925 | 0.450969 |
| 1.0 | 0.5 | 0.5 | 0.3 | 0.1 | 0.5 | 1.0 | 0.5 | 1.191841 | 0.113926 | 0.450972 |
| 1.0 | 0.5 | 0.5 | 0.3 | 0.1 | 0.5 | 0.1 | 0.0 | 1.273189 | 0.113787 | 0.449875 |
| 1.0 | 0.5 | 0.5 | 0.3 | 0.1 | 0.5 | 0.1 | 0.5 | 1.192154 | 0.113925 | 0.450966 |
| 1.0 | 0.5 | 0.5 | 0.3 | 0.1 | 0.5 | 0.1 | 1.0 | 1.111407 | 0.114063 | 0.452051 |

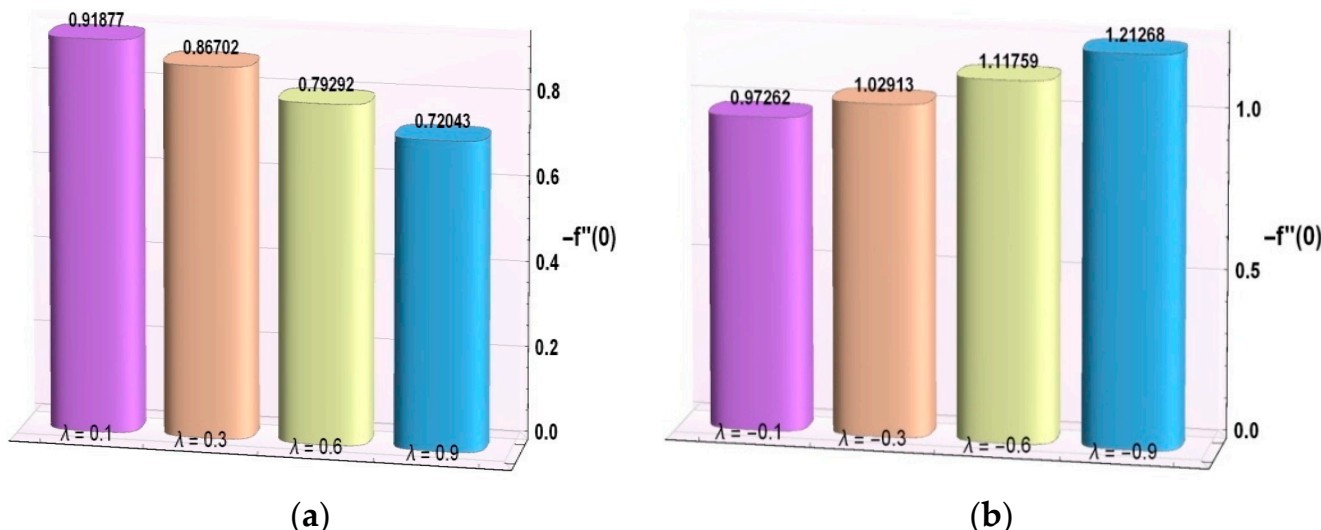

**Figure 5.** Behavior of (**a**)$-f''(0)$ for $\lambda \succ 0$ and (**b**)$-f''(0)$ for $\lambda \prec 0$.

**Table 2.** Numerical values of Sherwood number and density number against significant parameters.

| M | $N_t$ | $N_b$ | $N_r$ | $R_b$ | $S_c$ | $L_b$ | $\lambda$ | $-\Phi'(0)$ | $-\xi'(0)$ |
|---|---|---|---|---|---|---|---|---|---|
| 0.0 | 0.5 | 0.5 | 0.3 | 0.1 | 0.5 | 0.1 | 0.5 | 0.219173 | 0.211263 |
| 0.5 | 0.5 | 0.5 | 0.3 | 0.1 | 0.5 | 0.1 | 0.5 | 0.034016 | −0.195854 |
| 1.0 | 0.5 | 0.5 | 0.3 | 0.1 | 0.5 | 0.1 | 0.5 | 0.221225 | 0.209235 |
| 1.5 | 0.5 | 0.5 | 0.3 | 0.1 | 0.5 | 0.1 | 0.5 | 0.226416 | 0.208572 |
| 2.0 | 0.5 | 0.5 | 0.3 | 0.1 | 0.5 | 0.1 | 0.5 | 0.232671 | 0.208046 |
| 1.0 | 0.0 | 0.5 | 0.3 | 0.1 | 0.5 | 0.1 | 0.5 | 0.298319 | 0.228614 |
| 1.0 | 0.3 | 0.5 | 0.3 | 0.1 | 0.5 | 0.1 | 0.5 | 0.246424 | 0.215784 |
| 1.0 | 0.6 | 0.5 | 0.3 | 0.1 | 0.5 | 0.1 | 0.5 | 0.211511 | 0.206897 |
| 1.0 | 1.0 | 0.5 | 0.3 | 0.1 | 0.5 | 0.1 | 0.5 | 0.191963 | 0.205424 |
| 1.0 | 0.5 | 0.3 | 0.3 | 0.1 | 0.5 | 0.1 | 0.5 | 0.119955 | 0.159634 |
| 1.0 | 0.5 | 0.6 | 0.3 | 0.1 | 0.5 | 0.1 | 0.5 | 1.208556 | 0.916624 |
| 1.0 | 0.5 | 1.0 | 0.3 | 0.1 | 0.5 | 0.1 | 0.5 | 1.123307 | 0.878577 |
| 1.0 | 0.5 | 0.5 | 0.0 | 0.1 | 0.5 | 0.1 | 0.5 | 1.246378 | 0.933077 |
| 1.0 | 0.5 | 0.5 | 0.5 | 0.1 | 0.5 | 0.1 | 0.5 | 1.246420 | 0.932866 |
| 1.0 | 0.5 | 0.5 | 1.0 | 0.1 | 0.5 | 0.1 | 0.5 | 1.246461 | 0.932654 |
| 1.0 | 0.5 | 0.5 | 0.3 | 0.0 | 0.5 | 0.1 | 0.5 | 1.246393 | 0.933000 |
| 1.0 | 0.5 | 0.5 | 0.3 | 0.5 | 0.5 | 0.1 | 0.5 | 1.246445 | 0.932750 |
| 1.0 | 0.5 | 0.5 | 0.3 | 1.0 | 0.5 | 0.1 | 0.5 | 1.246497 | 0.932499 |
| 1.0 | 0.5 | 0.5 | 0.3 | 0.1 | 0.0 | 0.1 | 0.5 | 1.194672 | 0.916820 |
| 1.0 | 0.5 | 0.5 | 0.3 | 0.1 | 0.5 | 0.1 | 0.5 | 1.246403 | 0.932950 |
| 1.0 | 0.5 | 0.5 | 0.3 | 0.1 | 1.0 | 0.1 | 0.5 | 1.298610 | 0.949438 |
| 1.0 | 0.5 | 0.5 | 0.3 | 0.1 | 0.5 | 0.0 | 0.5 | 1.246403 | 0.923494 |
| 1.0 | 0.5 | 0.5 | 0.3 | 0.1 | 0.5 | 0.5 | 0.5 | 1.246403 | 0.971122 |
| 1.0 | 0.5 | 0.5 | 0.3 | 0.1 | 0.5 | 1.0 | 0.5 | 1.246403 | 1.019555 |
| 1.0 | 0.5 | 0.5 | 0.3 | 0.1 | 0.5 | 0.1 | 0.0 | 1.246504 | 0.932512 |
| 1.0 | 0.5 | 0.5 | 0.3 | 0.1 | 0.5 | 0.1 | 0.5 | 1.246403 | 0.932950 |
| 1.0 | 0.5 | 0.5 | 0.3 | 0.1 | 0.5 | 0.1 | 1.0 | 1.246302 | 0.933385 |

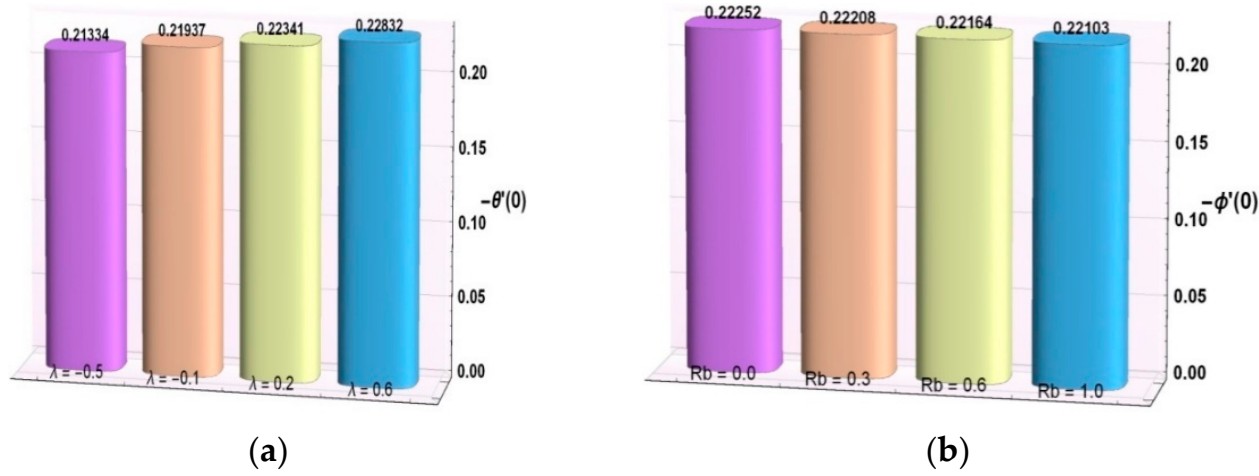

**Figure 6.** Behavior of (**a**) $-\theta'(0)$ for $\lambda$ and (**b**) $-\phi'(0)$ for Rb.

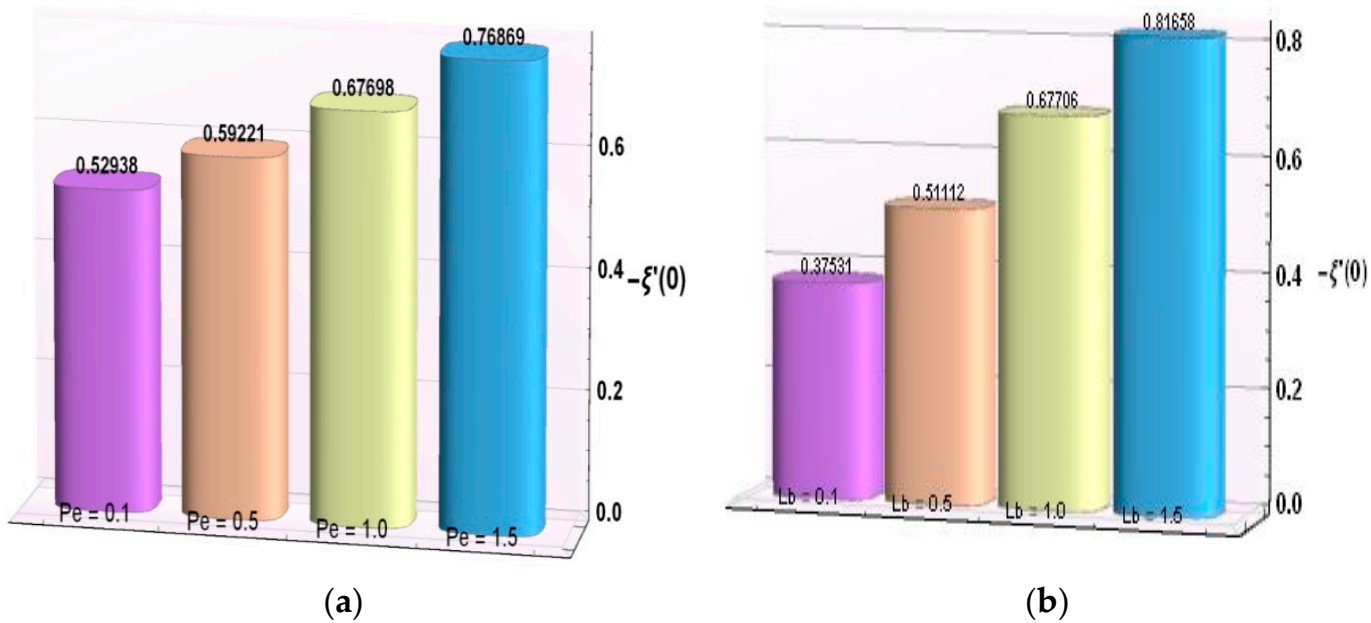

**Figure 7.** Behavior of (**a**) $-\xi'(0)$ for Pe and (**b**) $-\xi'(0)$ for Lb.

## 5. Conclusions

Bioconvection transport of Newtonian fluid encompassing gyrotactic microorganisms in three dimensions due to bidirectional expansion of sheet is numerically investigated by adopting two different numerical techniques. Assisting and opposing flow situations are considered with the effects of variable transport properties. The behavior of flow fields as well as physical quantities against inserting parameters is examined and the major outcomes are summarized as follows:

For controlling parameters regarding variable transport properties including $c_2$, $c_4$, $c_6$ and $c_8$, the velocity, temperature, concentration, and bioconvection density distributions accelerates, respectively. This fact may be beneficial in heat and mass transfer processes.

Error analysis of flow variables shows an absolute error of negligible magnitude, which guarantees the convergence of solutions.

Magnetic parameter decelerates the velocity field. Further, rate of heat transfer and density number decrease but surface friction and concentration rate grow up.

Stretching ratio parameter enlarges the vertical velocity component.

Temperature of nanofluid enhances against Brownian motion parameter while a conflicting behavior is observed for density field against Peclet number.

The estimated results of substantial quantities are presented via bar charts for a range of parameters and depicted that skin friction coefficient has reverse trend for both assisting and opposing flow conditions.

Heat transfer rate increases in assisting flow situations, while mass transfer rate depressed for bioconvection Rayleigh number. Further, density number amplifies for Peclet number as well as bioconvection Lewis number.

In future, one may investigate in modern intelligent computing-based stochastic solvers [45–50] for numerical solutions of the model representing the magneto-hydrodynamic three-dimensional bioconvection rheology of nanomaterial involving gyrotactic microparticles and similar studies.

**Author Contributions:** M.A.: Conceptualization, Methodology, writing—original draft preparation; S.E.A.: Methodology, Investigation, writing—review and editing; M.A.Z.R.: Software, Validation; N.P.: Resources, Formal analysis; W.U.K.: Data curation, Visualization; M.Y.M.: Resources, Supervision; Y.H.: Project administration, Funding acquisition. All authors have read and agreed to the published version of the manuscript.

**Funding:** This work was supported by the National Natural Science Foundation of China under Grant No. 51977153 and 51577046; the State Key Program of National Natural Science Foundation of China under Grant No. 51637004; the National Key Research and Development Plan (China) "important scientific instruments and equipment development" under Grant No. 2016YFF0102200; and Equipment research project in advance (China) under Grant No. 41402040301.

**Institutional Review Board Statement:** Not applicable.

**Informed Consent Statement:** Not applicable.

**Data Availability Statement:** Not applicable.

**Conflicts of Interest:** On behalf of all authors, the corresponding author states that there is no conflict of interest.

## Abbreviations

| | | | |
|---|---|---|---|
| $b$ | Chemotaxis constant (m) | $L$ | Characteristic Length (m) |
| $\overline{u_w}(\overline{x})$ | Velocity Sheet in x dimension (ms$^{-1}$) | $\alpha$ | Stretching ratio parameter |
| $\overline{v_w}(\overline{y})$ | Velocity Sheet in y dimension (ms$^{-1}$) | $Nb$ | Brownian motion parameter |
| $\mu_\infty$ | Constant dynamic viscosity (kg/m.s) | $Lb$ | Bioconvection Lewis number |
| $k_\infty$ | Constant thermal conductivity (m.kg/s$^3$.K) | $\theta(\eta)$ | Temperature profile |
| $f'(\eta)$ | x-component of Velocity field | $g'(\eta)$ | y-component of velocity field |
| $\mu(C)$ | Variable dynamic viscosity (kg/m.s)) | $Nt$ | Thermophoretic parameters |
| $T_w$ | Surface temperature (K) | $\tau_w$ | Skin friction coefficient (kg/s$^2$.m) |
| $T_\infty$ | Ambient temperature (K) | $Rb$ | Bioconvection Rayleigh number |
| $M$ | Magnetic parameter | $C_{f_{\overline{x}}}$ | Skin friction along $x$-axis |
| $c_p$ | Heat capacity at constant pressure | $C_{f_{\overline{y}}}$ | Skin friction along $y$-axis |
| $B$ | Volumetric expansion coefficient of nanofluid (1/K) | $k(C)$ | Variable Thermal conductivity (m.kg/s$^3$.K) |
| $W_c$ | Maximum swimming speed (m/s) | $D_B(C)$ | Diffusivity of Brownian motion |
| $c_2$ | Dimensionless viscosity | $D_n(C)$ | Diffusivity of microorganism |
| $c_4$ | Thermal conductivity | $D_T$ | Thermophoretic diffusion coefficient |
| $c_6$ | Mass diffusivity | $D_{n,\infty}$ | Constant microorganism diffusivity |
| $c_8$ | Microorganism diffusivity | $D_{B,\infty}$ | Constant mass diffusivity |
| $q_w$ | Surface heat flux coefficient (kg/s$^3$) | $Nu_{\overline{x}}$ | Nusselt number |
| $q_m$ | Surface mass flux coefficient (m/s) | $Sh_{\overline{x}}$ | Sherwood number |
| $q_n$ | Surface motile microorganism flux (m/s) | $Nn_{\overline{x}}$ | Density number of motile microorganisms |
| $\phi(\eta)$ | Concentration profile | $\gamma$ | Density of motile microorganisms |
| $\xi(\eta)$ | Density profile | $\tau'$ | Ratio of heat capacity of nanoparticles to heat capacity of fluid |
| $C_w$ | Nanoparticles concentration at wall | $C_\infty$ | Nanoparticles concentration at free space |
| $Re$ | Reynold's number | $B_0$ | Magnetic field strength (kg/s$^2$.m$^2$) |
| $n_w$ | Density of microorganisms at wall | $n_\infty$ | Ambient density of microorganisms |
| $\rho_\infty$ | Ambient density of fluid (kg/m$^3$) | $\rho_p$ | Density of nanoparticles (kg/m$^3$) |
| $\rho_m$ | Density of microorganisms (kg/m$^3$) | $\sigma$ | Electric conductivity (s$^3$.m$^2$/kg) |
| $\eta$ | Similarity variable | $\nu_\infty$ | Kinematic viscosity at free space (m$^2$/s) |

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
