# Peer review of "Effects of Variable Transport Properties on Heat and Mass Transfer in MHD Bioconvective Nanofluid Rheology with Gyrotactic Microorganisms: Numerical Approach"

_coatings, doi:10.3390/coatings11020231_

Round 1

Reviewer 1 Report

In this work, authors have presented a numerical analysis of flow dynamics of the bio-convection phenomenon involving gyrotactic micro-particles. Authors need to address the following points :

  1. The title of the manuscript does not accurately represent the work. It needs to include the term numerical analysis since that is what the manuscript covers. Also, the work focuses on micro-particles so the use of the term 'micro-organisms' in the title is misleading. 
  2. The abstract needs to be re-written to discuss the novelty of the work (specific novelty). Authors have summarized what they have done, however, the impact and significance is not clear. Authors have mentioned the controlling parameters in the abstract, however it is not very clear what controlling parameters refer to. (Details mentioned later in the manuscript cannot be included in the abstract unless they are self-explanatory).
  3. Introduction needs to be re-written.
    1. Line 29, please specify what authors are referring to as 'upper part' of? 
    2. Line 31: what do authors mean by 'massive'? vague qualifiers should be avoided when writing scientific articles.
    3. Line 32: please provide a more specific terminology like 'increased density' instead of 'stuffed' and 'unbalanced'? vague qualifiers should be avoided.
    4. Authors have often used / instead of and. '/' refers to 'or' and is not appropriate use in a number of places it has been used in the manuscript.
    5. The Introduction does not clearly discuss the motivation, significance and impact of the work.
    6. Novelty of the work is not clear and it's direct applications also need to be discussed.
    7. Although authors have briefly summarized some previous work, it has not been discussed in a contiguous manner to lead to the work that the authors have presented. Authors should discuss the previous work in a little more detail.
    8. Authors keep mentioning that their work is novel, however, the novelty of the work is unclear.
  4. The equations need to be aligned and the symbols in the text need to be aligned with their respective parameters. For example: see line 89.
  5. The boundary conditions and the associated assumptions need to be mentioned and discussed.
  6. The skin friction drag needs to be discussed as well.
  7. Authors have not appropriately discussed the plots. For example, authors have not even mentioned that Pe is Peclet number ( even though it is common knowledge, but it should be mentioned). The aplot has not been discussed properly.
  8. Fig.2-7 can be combined into a single figure with multiple panels. Convergence plots can also be plotted to show changes in absolute error.
  9. Significance of why each parameter is being evaluated should be discussed. For example Peclet number is related to the motion of cells. Similarly, each evaluated parameter should be described in its context to the application.
  10. Figs. 8-15 can be combined into a single figure with multiple panels.
  11. There is very little discussion of the results. Authors need to elaborate on each of the plots and what they can interpret from it.
  12. Novelty of the work is not clear. It is very similar to the work presented by Nadeem et al. " Mathematical analysis of bio-convective micropolar nanofluid", Journal of Computational Design and Engineering 6 (2019) 233–242.                                                                                     Please proof-read the entire manuscript. Overall, the manuscript needs a lot of work.

Author Response

Reviewer 1 General Comments

In this work, authors have presented a numerical analysis of flow dynamics of the bio-convection phenomenon involving gyrotactic micro-particles. Authors need to address the following points :

Author Response:

We are thankful for the constructive comments regarding improvement to our work. We have revised the manuscript accordingly and point to point responses are provided here.

Reviewer 1 Query 1:

  1. 1: The title of the manuscript does not accurately represent the work. It needs to include the term numerical analysis since that is what the manuscript covers. Also, the work focuses on microparticles so the use of the term 'micro-organisms' in the title is misleading.

Author Response

According to suggestion, the title of the manuscript is rewritten which completely describes the work as suggested. Moreover, we have studied nanofluid containing microorganism, therefor this term is included.

Previous Title : Bio-nanofluid Convective Rheology with Variable Transport Characteristics in the Presence of Micro-Organisms

New Title: Effects of Variable Transport Properties on Heat and Mass Transfer in MHD Bio-convective Nanofluid Rheology with Gyrotactic Microorganisms: Numerical Approach

Please see the title in revised manuscript.

Reviewer 1 Query 2:

The abstract needs to be re-written to discuss the novelty of the work (specific novelty). Authors have summarized what they have done, however, the impact and significance is not clear. Authors have mentioned the controlling parameters in the abstract, however it is not very clear what controlling parameters refer to. (Details mentioned later in the manuscript cannot be included in the abstract unless they are self-explanatory).

Author Response:

Agreed. We have updated the manuscript by modifying the abstract as directed. Additionally, contribution  and innovative insights are further highlighted in the introduction section as follows.

  • A novel investigation has been presented for bio-convection phenomenon involving gyro-tactic micro-organisms in three dimensional flow dynamics of nanofluid.
  • Variable transport properties, assisting and opposing flow situations combined with magnetic field properties are incorporated in the model.
  • Mathematical modeling is performed by utilization of the conservation laws of mass, momentum, energy, mass fraction and bio-convection processes along with suitable scaling procedure for the construction of system of differential equations.
  • Lie group analysis approach is presented to compute the absolute invariants for the differential system along with error analysis for validation of computed results.
  • Graphical and numerical illustrations are prepared in order to present the physical insight of the considered analysis.

Please see highlighted text in the “Abstract” and “Introduction” Section of revised manuscript.  

Reviewer 1 Query 3:

Introduction needs to be re-written.

3(1). Line 29, please specify what authors are referring to as 'upper part' of?

Author Response: Introduction section is improved and additional information for microorganisms along with applications are now discussed elaboratively . Moreover, the pointed correction also addressed in the revised manuscript.

Please see the  paragraph 2 of “Introduction” Section.

3(2). Line 31: what do authors mean by 'massive'? vague qualifiers should be avoided when writing scientific articles.

Author Response :. This term is removed. Please check the introduction section

3(3). Line 32: please provide a more specific terminology like 'increased density' instead of 'stuffed' and 'unbalanced'? vague qualifiers should be avoided.

Author Response: In revised version, we have described the terminologies carefully.

3(4). Authors have often used / instead of and. '/' refers to 'or' and is not appropriate use in a number of places it has been used in the manuscript.

Author Response: Please see revised version of manuscript. Such symbols are removed.

3(5). The Introduction does not clearly discuss the motivation, significance and impact of the work

Author Response: As suggested, we have carefully improved introduction by keeping all these factors about present work. Additionally, contribution  and innovative insights are further highlighted in the introduction section as follows.

  • A novel investigation has been presented for bio-convection phenomenon involving gyro-tactic micro-organisms in three dimensional flow dynamics of nanofluid.
  • Variable transport properties, assisting and opposing flow situations combined with magnetic field properties are incorporated in the model.
  • Mathematical modeling is performed by utilization of the conservation laws of mass, momentum, energy, mass fraction and bio-convection processes along with suitable scaling procedure for the construction of system of differential equations.
  • Lie group analysis approach is presented to compute the absolute invariants for the differential system along with error analysis for validation of computed results.
  • Graphical and numerical illustrations are prepared in order to present the physical insight of the considered analysis.

Please see highlighted text in the “Introduction” Section of revised manuscript. 

3(6). Novelty of the work is not clear and it's direct applications also need to be discussed.

Author Response: In the “Abstract”, “Introduction” and “Conclusions” sections of revised manuscript, novelty of this work is more elaborately presented for better understanding of readers.

3(7). Although authors have briefly summarized some previous work, it has not been discussed in a contiguous manner to lead to the work that the authors have presented. Authors should discuss the previous work in a little more detail.

Author Response: We have rewritten the “Introduction” Section by including the information for previous work on the topic presented and latest state of art on the subject matter, as suggested.

3(8). Authors keep mentioning that their work is novel, however, the novelty of the work is unclear.

Author Response: We have presented the novelty of the given study more elaborately, please see, innovative contributions and insights in the “Introduction” Section of the revised manuscript.

Reviewer 1 Query 4:

The equations need to be aligned and the symbols in the text need to be aligned with their respective parameters. For example: see line 89.

Author Response:

As recommended, section 3 is carefully reviewed and corrected for equations alignment and symbols in the text.

Reviewer 1 Query 5:

The boundary conditions and the associated assumptions need to be mentioned and discussed. 

 Author Response:

In the revised manuscript, mathematical model is thoroughly described along with associated boundary conditions for better understanding of the readers as suggested.

Reviewer 1 Query 6:

The skin friction drag needs to be discussed as well.

Author Response:

In the revised version, skin friction co efficient is discussed with its physical description as suggested.

Reviewer 1 Query 7:

Authors have not appropriately discussed the plots. For example, authors have not even mentioned that Pe is Peclet number ( even though it is common knowledge, but it should be mentioned). The aplot has not been discussed properly.

Author Response:

According to your valuable recommendation, numerical and graphical illustration of system model are exhaustively discussed, and the description of each parameter is provided.

Please see the highlighted text in “Results and Discussion” section.

Reviewer 1 Query 8:

Fig.2-7 can be combined into a single figure with multiple panels. Convergence plots can also be plotted to show changes in absolute error.

Author Response:

As suggested, figures are rearranged with multiple panels. Convergence of the technique is confirmed with the help of error analysis.

Reviewer 1 Query 9:

Significance of why each parameter is being evaluated should be discussed. For example Peclet number is related to the motion of cells. Similarly, each evaluated parameter should be described in its context to the application.

Author Response:

In the revised version of the manuscript, influence of each parameter of interest is explained evidently according to its application as suggested.

 Please see the highlighted text in “Results and Discussion” section.

Reviewer 1 Query 10:

Figs. 8-15 can be combined into a single figure with multiple panels.

Author Response:

As recommended, figures are rearranged with multiple panels.

Please see the graphical illustration in “Results and Discussion” section.

Reviewer 1 Query 11:

There is very little discussion of the results. Authors need to elaborate on each of the plots and what they can interpret from it.

Author Response:

 In the revised version of the manuscript, the results of the problem are elaboratively presented with necessary discussion on physical significance as directed.

Reviewer 1 Query 12:

Novelty of the work is not clear. It is very similar to the work presented by Nadeem et al. " Mathematical analysis of bioconvective micropolar nanofluid", Journal of Computational Design and Engineering 6 (2019) 233–242.

Author Response:

In the revised version of manuscript, abstract is rewritten to present novelty more elaborately in terms of the physical factors, fluid model and adopted solution technique. Additionally, contribution  and innovative insights are further highlighted in the introduction section. Furthermore, the problem formulation is narrated more evidently to understand the contribution of presented study with respect to published literature.

Reviewer 2 Report

Presented work is very interesting in terms of both description of the problem as well as involving mathematical modelling combined with numerical solution and further analysis. 

It should be emphasis that pointing out aims of the work as well as providing short description of paper sequence at the end of Introduction are essential for clarity and better understanding for reader.

However, there are following points in my opinion that should be improved:

  1. providing more detailed description of other authors works presented in Introduction
  2. Line 30 - "...heavy density stratification." - linguistic revision needed
  3. Line 31 - "...massive than water." - linguistic revision needed
  4. Line 35, 64, 72 - there should be no space after slash - please check other part of the text
  5. Line 185 - fig. 8-15 - provide complete legend of diagram, e.g. fig 8. shows results obtain for different values of c2. However results are not assigned directly to different c2 values on chart

Author Response

Reviewer 2 General Comments

Presented work is very interesting in terms of both description of the problem as well as involving mathematical modelling combined with numerical solution and further analysis.

It should be emphasis that pointing out aims of the work as well as providing short description of paper sequence at the end of Introduction are essential for clarity and better understanding for reader.

There are following points in my opinion that should be improved:

Author Response:

Many thanks for your valuable comments “Presented work is very interesting in terms of both description of the problem as well as involving mathematical modelling combined with numerical solution and further analysis” on our submitted manuscript. Additionally, authors tried their best to address each query raised by the anonymous reviewer and manuscript is modified accordingly.

 Reviewer 2 Query 1:

providing more detailed description of other authors works presented in Introduction.

Author Response:

In the introduction section of the revised version, Literature review of this manuscript is improved with a broader discussion on the presented topic and latest state of the art.

Please see the introduction section of revised manuscript.

Reviewer 2 Query 2: Line 30 - "...heavy density stratification." - linguistic revision needed.

Author Response:

As suggested, the terminology of presented subject terms is provided professionally for better understanding of the readers. Additionally, the linguistic quality is improved considerably by avoiding grammatical errors and typos.

Reviewer 2 Query 3:. Line 31 - "...massive than water." - linguistic revision needed.

Author Response:

As directed, the terminology of presented subject terms is provided professionally for better understanding of the readers. Additionally, the linguistic quality is improved considerably by avoiding grammatical errors and typos.

Reviewer 2 Query 4:

Line 35, 64, 72 - there should be no space after slash - please check other part of the text.

Author Response:

In the revised version of the manuscript, the pointed corrections are rectified . Additionally, the linguistic quality is improved considerably by avoiding grammatical errors and typos

Reviewer 2 Query 5:

Line 185 - fig. 8-15 - provide complete legend of diagram, e.g. fig 8. shows results obtain for different values of c2. However, results are not assigned directly to different c2 values on chart.

Author Response:

As suggested, we have provided the legends of all figures in professional manner. Please see the legends of all graphical illustrations  in the “Result and Discussion” Section.

 Finally, the author thanks again the Editor in Chief, Associate Editor, Editorial Staff and the anonymous reviewer(s) for their time, support and efforts to improve the quality of the paper.

With best regards

Dr. Muhammad Awais

COMSATS University Islamabad, Attock Campus

Pakistan
